# Xanthanolides in *Xanthium* L.: Structures, Synthesis and Bioactivity

**DOI:** 10.3390/molecules27238136

**Published:** 2022-11-22

**Authors:** Jiaojiao Zhang, Rongmei Zhao, Lu Jin, Le Pan, Dongyu Lei

**Affiliations:** 1Department of Applied Chemistry, Chemistry and Chemical Engineering College, Xinjiang Agricultural University, Urumqi 830052, China; 2Institute for Drug Control of Xinjiang Uygur Autonomous Region, Urumqi 830054, China; 3Department of Physiology, Preclinical School, Xinjiang Medical University, Urumqi 830011, China

**Keywords:** xanthanolides, structures, synthesis, bioactivity

## Abstract

Xanthanolides were particularly characteristic of the genus *Xanthium*, which exhibited broad biological effects and have drawn much attention in pharmacological application. The review surveyed the structures and bioactivities of the xanthanolides in the genus *Xanthium*, and summarized the synthesis tactics of xanthanolides. The results indicated that over 30 naturally occurring xanthanolides have been isolated from the genus *Xanthium* in monomeric, dimeric and trimeric forms. The bioassay-guided fractionation studies suggested that the effective fractions on antitumor activities were mostly from weak polar solvents, and xanthatin (**1**) was the most effective and well-studied xanthanolide. The varieties of structures and structure-activity relationships of the xanthanolides had provided the promising skeleton for the further study. The review aimed at providing guidance for the efficient preparation and the potential prospects of the xanthanolides in the medicinal industry.

## 1. Introduction

The plants belonged to the genus *Xanthium* L. were found world wide, which showed important applications in traditional herbal medicine in many countries [1]. They were commonly used to treat nasitis, fever, arthritis, tumors, gastric ulcer and microbial infections [2]. In China, *Xanthium sibiricum* Patrin ex Widder was listed in the Pharmacopoeia for treating rhinitis, headache, rheumatism and many other diseases. However, the acute toxicity of *Xanthium* L. was also reported during the medicinal application and ingestion poisoning of live-stock [3]. The constituents of *X*. *strumarium* were responsible for the activity or toxicity of the species. To date, varieties of compounds have been isolated from the plants belonged to the genus *Xanthium*, including sesquiterpenoids, flavonoids, coumarins, steroids, phenylpropenoids, lignanoids, glycosides, anthraquinones, and naphthoquinones [1,4]. Some of these compounds had been proven to exhibit significant medicinal properties, such as diuretic, anthelmintic, antifungal, anti-inflammatory, antidiabetic, anticancer, and many other activities [5,6,7,8].

Xanthanolides were the characteristic ingredients of the genus *Xanthium*, which possess bicyclic sesquiterpene lactones with structural feature of five-membered γ-butyrolactone ring fused heptatomic carbocycle. The genus Xanthium is the richest in the limited plant species for the source of xanthanolides. Xanthanolides were reported to show various bioactivities, including antitumor, antifungal, anti-ulcerogenic, anti-ulcerogenic and insecticidal activity [5]. In recent years, xanthanolides have drawn increasing attention due to its promising skeleton and potential value [6,7,8,9,10]. In this paper, the structures, synthesis and bioactivities of the xanthanolides in the genus *Xanthium* L. were summarized for these unique structure’s potential use in the pharmaceutical industry.

## 2. Structures of Xanthanolides in Genus Xanthium

Xanthanolides were found in the genus Xanthium as the distinctive sesquiterpenes. Since the first isolation of xanthatin (**1**) [10] and xanthinin (**14**) [11] by Little et al., and Geissman et al., from the aerial parts of *Xanthium pennsylvanicum* in 1950s, over thirty xanthanolides have been isolated from the plant of the genus *Xanthium* in monomeric, dimeric or even trimeric forms. The identified natural xanthanolides from the genus *Xanthium* were listed in Table 1 and their structures were showed in Figure 1, indicating that the xanthanolides had the skeletal structures of guaiane or suaiane lactones, some of which were unsaturated at C-12 to form a α-methylene-γ-lactone ring. The structural diversities of the xanthanolides mainly stemmed from the side chain of C-1, the oxygen functions of C-11 and the stereochemistry of C-7 and C-8. The C-1 side chain mostly contained 4 carbons, which could be unsaturated to form α, β-unsaturated ketones (compound **1**~**11**), saturated to form hydroxy groups (**24** and **25**) or cyclized at C-4/C-5 (**34** and **35**). The The side chain could also be further bio-modified to form esters or glycosides by reacting with carboxylic acid and glucose (compounds **26**, **27**, **30** and **31**). The methylene-γ-lactone ring and the side chain of C-4 were reported to be critical for the activity of xanthanolides. 

The genus Xanthium was the richest source of various xanthanolides. Xanthatin (**1**) and xanthinin (**14**), as the firstly isolated xanthanolides, were then found to be commonly existed in various *Xanthium* species with many other xanthanolides by different groups. Up to now, Xanthatin (**1**) [14,15,16,17,18,19,20,21,22,23,24] has been identified from *Xanthium cavanillesii*, *Xanthium italicum*, *Xanthium macrocarpum*, *Xanthium macrocarpum*, *Xanthium orientale*, *Xanthium pennsylvanicum* and *Xanthium spinosum.* 8-epi-xanthatin (**2**) [19,25,26,27,28,29,30,31,32,33] were found in *Xanthium pungens*, *Xanthium spinosum*, *Xanthium strumarium*, *Xanthium brasilicum* and *Xanthium indicum*. Bohlmann et al. [18] and Abou-Elzahab et al. [23] isolated xanthatin-1β,5β-epoxide (**3**) from the *Xanthium spinosum* seperately. 8-epi-xanthatin-1β,5β-epoxide (**4**) [26,28,30,32,33,34,35,36,37,38] was isolated from *Xanthium pungens*, *Xanthium strumarium* and *Xanthium brasilicum* seperately. In 1994, Mondal et al., isolated 6β,9β-Dihydroxy-8-epi-xanthatin (**5**) [39] from the *Xanthium strumarium.* Xanthatin-1α,5α-epoxide (**6**) [14,18,40] could be isolated from the *Xanthium spinosum* and *Xanthium strumarium.* Finzi et al. [41] isolated 11a,13-Dihydro-8-epi-xanthatin (**7**) from the whole plant of *Xanthium catharticum.* 11α,13-dihydroxanthatin (**9**) [19] was isolated from the aerial parts of *Xanthium strumarium*. 11a,13-dihydro-8-epi-xanthatin-1a,5a-epoxide (**10**) and 11a,13-dihydro-8-epi-xanthatin-1b,5b-epoxide (**11**) [41] were isolated from *Xanthium catharticum* in 1991. In 1997, Joshi et al., isolated tomentosin (**12**), 8-epi-xanthatin-1β,5β-epoxide (**4**), xanthumin (**20**) and 8-epi-xanthatin (**2**) from *X*. *strumarium* [26]. Tomentosin (**12**) was commonly found in *Xanthium indicum* and *Xanthium pungens* [42,43,44,45,46,47,48,49,50,51]. 4-epi-xanthanol (**13**) was isolated from *Xanthium italicum*, *Xanthium macrocarpum*, *Xanthium macrocarpum* and *Xanthium strumarium* [14,20,21,24,52]. Xanthinin (**14**) had been commonly found in *Xanthium italicum*, *Xanthium macrocarpum*, *Xanthium orientale*, *Xanthium pennsylvanicum*, *Xanthium spinosum* and *Xanthium strumarium* [10,11,15,21,22,23,40,53,54,55,56]. In 2005, Hasegawal et al., isolated the fruits of *Xanthium macrocarpum* and obtained seven xanthanolides including xanthinosin (**22**), xanthatin (**1**), 4-hydroxyxanthinosin (**24**), xanthinin (**14**), 4-epixanthanol (**16**), 4-epiisoxanthanol (**17**), 2-hydroxyxanthinosin (**23**). In 1990, two dimeric xanthanolides, pungiolide A (**35**) and pungiolide B (**36**), were firstly isolated from the *Xanthium pungens* by Ahmed et al., Meanwhile, many other xanthanolides, including 8-*epi*-xanthatin [29], tomentosin (**12**), 2-hydroxytomentosin (**23**),xanthumin (**20**)and 8-*epi*-xanthatin-1β,5β- epoxide (**4**) were also been isolated [19]. In 2021, three xanthanolide sesquiterpene trimers, Xanthanoltrimer A, B and C (**38**, **39** and **40**) were firstly isolated from the fruits of *Xanthium italicum Moretti* by Yu et al., however they were all found to show no significant biologic activity [67]. The details on the sources and isolation of xanthanolides regarding the sources and isolation were listed in the Table 1.

## 3. The Biological Activity of Xanthanolides

*X. strumarium* has been used for a long time in clinic as the traditional medicine due to its broad activity. The fruits of *X. strumarium*, named as “Cang-Er-Zi” in China, have been used to treat rhinitis, anti-inflammory and many other diseases. Xanthanolides, as the typical constitutes of the genus *Xanthium*, showed important medicinal values by exhibiting a broad spectrum of biological effects, including anti-tumour, antimalarial, antifungal, antiviral and anti-inflammatory activities.

### 3.1. Anti-Tumor Activity

The anti-tumor activities of some *Xanthium* species have been discovered for a long history, such as *X. canadense*, *X. catharticum*, *X. chinense*, *X. echinatum*, *X. indicum*, *X. italicum*, *X. macrocarpum*, *X. orientale*, *X. spinosum* and *X. strumarium*. [5,20] In our previous study, the leaf extract of *X. italicum* was found to exhibit significant cytotoxic activity against human tumour cell lines A549 and hep G2 [68]. To date, the crude extracts of many *Xanthium* species, including *X. strumarium* L., *X. italicum* and et al., exhibited antitumor acitivity. The bioassay-directed fractionation and cytotoxicity investigation revealed that the weak polar solvents, such as dichloro-methane or chloroform, were the most effective fractions. Our previous study conducted a bioassay-directed isolation of the crude extract of *X. italicum*, which led to the identification of two effective xanthanolides, xanthatin (**1**) and xanthinosin (**22**), from the chloroform fraction [68].

In 2002, Roussakis et al., conducted bioassay-directed fractionation of the dichloromethane fraction of *Xanthium strumarium*, which led to the isolation of xanthatin (**1**) [31]. Therefore, it could be concluded that xanthatin (**1**) was mainly responsible for the cytotoxicity of *Xanthium* species. To date, xanthatin (**1**) has been proven to exhibit high activity against many human tumors by many teams, including the human bronchial epidermoid carcinoma NSCLC-N6 with IC_50_ of 3 ug/mL [69]. Kim et al., isolated 8-epi-Xanthatin (**2**) and 8-epi-xanthatin-1b,5b-epoxide (**4**) from *Xanthium strumarium* and found they demonstrated significant inhibition on the non-small-cell lung A549, SK-OV-3 (ovary), SK-MEL-2, HCT-15 and XF498 (CNS) cell lines in vitro [32]. The structure-activity relationships study revealed that the activity decreased significantly when the olefinic bond at C1-C5 of 8-epi-Xanthatin (**2**) was oxidized to form 8-epi-xanthatin-1β, 5β-epoxide (**4**) [5]. In addition, the activities were all decreased sharply for three cell lines when the C4 ketone of the tomentosin (**12**) was hydrogenized to form 4-dihydrotomentosin (**24**) [34]. In 2009, Kovacs et al., extracted the leaves of *X*. *italicum* with chloroform and investigated its antitumor activity against HeLa, A431 and MCF7 cells. The bioassay-guided fractionation led to the isolation of the active xanthanolides, xanthatin (**1**), 4-epi-isoxanthanol (**17**) and 2-hydroxyxanthinosin (**23**), and xanthatin (**1**) exerted the most significant inhibition effect on cell proliferation. [20].

As the most acknowledged antitumor xanthanolides, the mechanism studies on the cytotoxicity of xanthatin (**1**) have been intensively studied in recent years. In 2012, Lu et al., reported that xanthatin (**1**) exhibited significant antitumor activity against non-small-cell lung cancer cells A549 through cell cycle arrest and apoptosis by disrupting NF-kappa B signaling [70]. In the following study, they found that xanthatin (**1**) could suppress DNA replication by triggering Chk1-mediated DNA damage [71]. In 2018, Fang et al., reported that xanthatin (**1**) could promote the oxidative stress-mediated apoptosis of HeLa cells via inhibiting thioredoxin reductase, which was supported by the research of Zhang et al., in 2021 [72,73]. In 2019, Yu et al., demonstrated that xanthatin could covalently bind to JAK and IKK kinases and inhibit the STAT3 and NF-κB signalling pathways to suppress the development of cancer and inflammatory diseases [74]. In 2020, Feng et al., reported that xanthatin (**1**) could activate endoplasmic reticulum stress-dependent CHOP pathway to induce the apoptosis of glioma cell and inhibit the growth [75]. In 2021, Jia et al., reported that xanthatin could selectively inhibit the proliferation of retinoblastoma cells by inhibiting the PLK1- mediated cell cycle [76].

### 3.2. Antimicrobial Activity

The antimicrobial activity of the *Xanthium* species has been reported and applied for a long time. Rodino et al., reported the the antimicrobial activity of extracts from *Xanthium strumarium* against phytophthora infestans [77]. Tsankova et al., investigated the antibacterial activity of the leaf extract of *Xanthium italicum* and isolated two xanthanolides: xanthinin (**14**) and xanthatin (**1**). The results showed that both the crude extract and isolated xanthanolides all exhibited significant activities against the Gram-positive bacterium *Staphylococcus aureus* [15].

Lavault et al., studied the antileishmanial and antifungal activities of seven xanthanolides isolated from *Xanthium macrocarpum*, including xanthanolides, xanthinosin (**22**), xanthatin (**1**), 4-hydroxyxanthinosin (**24**), xanthinin (**14**), 4-epiisoxanthanol (**17**), 4-epixanthanol (**16**) and 2-hydroxyxanthinosin (**23**), the results showed that xanthatin and xanthinin exhibited significant fungistatic activities with MIC of 32 ug/mL. And five of the xanthanolides tested were found to be leishmanicidal, among which xanthinin (**14**) was proved to be the most active element [21].

Sato et al., isolated xanthatin (**1**) from the leaves of *Xanthium sibiricum* Patr er Widd and found that xanthatin could inhibit the growth of 20 strains of methicillin-resistant *Staphylococcus aureus* (MRSA) and 7 strains of methicillin-susceptible *Staphylococcus aureus* (MSSA) with the MICs of the range from 7.8 to 15.6 ug/mL. However, it showed no inhibitory effect on *Escherichia coli* [15].

Xanthatin (**1**) had been widely reported as the main antimicrobial component among all the xanthanolides, which had been recognized as an important precursor for the bio-mimic synthesis in exploring the novel antimicrobial agents. Zhi et al., synthesized a series of Michael type amino derivatives from xanthatin (**1**) and investigated their antifungal activities against several phytopathogenic fungi, and led to the finding of the more effective ompounds than xanthatin [78]. In 2009, Liu et al., reported that the water extracts of *X. strumarium* exhitbited antiviral activity against duck hepatitis B virus and xanthatin (**1**) exerted significant activity against influenza A virus [79]. Tsankova et al., found that the leaf extract exerted antiviral activity against the pseudorabies virus A-2 strain, while both xanthinin (**14**) and xanthatin (**1**) showed no antiviral activity. The further study showed that all the three samples above displayed no antiviral activity against influenza A/chicken/H7N7 (FPV) and Newcastle disease virus (NDV) [15]. The results indicated that the xanthanolides might have very weak anti-viral activity.

### 3.3. Anti-Inflammatory Activity

*X. strumarium* has long been used to inhibit various inflammatory diseases and its crude extracts, either water or methanol, have been confirmed to show effects via decreasing IFN-γ, NO production and TNF-α production. In 2005, Kim et al., investigated the anti-inflammatory activities of the methanol extracts of *X. strumarium*, which indicated that the crude extract could down-regulate the mRNA express of NO, PGE 2 and TNF-α [80]. In 2008, Yoon et al., treated the cells with the ethyl acetate extract of X. strumarium, which was found to reduce the production of NO to 0.9 uM. With the further purification, xanthatin (**1**) and xanthinosin (**22**) were obtained and were proven to inhibit the production of NO and activated the microglia in a dose-dependent manner (IC_50_ = 0.47, 11.2 and 136.5 mM, respectively) [12]. In 2022, Liu et al., investigated the anti-inflammatory effect of xanthatin (**1**) and found that xanthatin (**1**) could decrease the amount of nitric oxide (NO), reactive oxygen species (ROS) and down regulate the expression of inflammatory cytokines including COX-2, TNF-α, IL-β, and IL-6 and et al. The results demonstrated that xanthatin (**1**) could exhibit anti-inflammatory effects by down regulating NF-kappaB, MAPK and STATs signaling pathways [81].

Li et al., reported that isoxanthanol (**17**) could exhibited protective and anti-inflammatory effects on subchondral bone deterioration. The results indicated that isoxanthanol could inhibit the excessive release of interleukin-6, NO and PGE2 in a dose dependent manner both in vitro and in vivo [82].

## 4. Synthesis of Xanthanolides

Xanthanolides were the characteristic sesquiterpenoids from the genus Xanthium. The synthesis of xanthanolides are drawing increasing attention because of the unique structure, impressive biological profile and low content in plant.

### 4.1. Chemical Synthesis

The seven-membered carbocycle fused γ-butyrolactone was the structural core of xanthanolides and the main difficulty to overcome. In 2003, Nosse et al., prepared the 5,7-fused xanthanolide core via the cyclopropanation of furan-2-carboxylic methyl ester [22]. In 2004, Vaissermann et al., prepared the 5,7-fused skeleton of xanthanolides via complexes of cycloheptatriene carboxylic acid esters and chromium tricarbonyl in three steps [83]. In 2021, Tang et al., reported the efficient synthesis of the α-methylene- β-lactones efficiently via dyotropic rearrangement, which realized the preparation of the building blocks of natrual xanthanolides [84].

The first total synthesized xanthanolide was 11, 13-dihydroxanthatin lacking the reactive exocyclic methylene reported by Morken et al., in 2005 [85]. In the same year, Martin et al., reported the first total synthesis of the (+)-8-epi-xanthatin starting from the commercially available ester in 14 steps with overall yield of 5.5% [86]. In 2008, Shishido et al., completed the enantioselective total synthesis of xanthatin by the construction of the vinyl functionality at C1 and the exo-methylene at C11 via the syn-elimination of selenoxides from an optically pure cis-fused bicyclic lactone, and then, they reported an enantioselective total synthesis of 8-epi-xanthatin (**2**) by developing a synthetic route without the stoichiometric selenium reagents in 2012 [87,88]. In 2012, Tang et al., developed a method to synthesize the enantiopure γ-butyrolactones via a controllable Wagner–Meerwein-type dyotropic rearrangement of cis-β-lactones and successfully applied it to prepare a number of xanthanolides [89]. In 2013, Kenji et al., reported the enantioselective total syntheses of xanthatin and 11, 13-dihydroxanthatin via stereocontrolled conjugate allylation to an optically pure γ-butenolide [90]. In 2017, Tang et al., improved the previous method and completed the synthesis of (+)-8-epi-xanthatin (**2**) through a CPA-catalyzed tandem allylboration/lactonization reaction and prepared a series of xanthanolides efficiently [9]. The representative tactics on the total synthesis of xanthanolides were shown in Figure 1.

### 4.2. Biological Synthesis

Synthetic biology was proposed to be an alternative strategy to produce sesquiterpenoids and some outstanding achievements has been made in this field [91]. The bio-synthetic strategy requires the fully identification of xanthanolides related metabolite genes of the plants and revelation of the enzymes involved in the biosynthesis. The skeletons of xanthanolides were proposed to be formed by the modifications on the backbone of germacranolide [5]. In recent years, the biogenesis of xanthanolides has been elucidated from the plant species. In 2010, a sesquiterpene synthase (STP) forming δ-guaiene was isolated from cultured cells of Aquilaria, which enabled the conversion of the common substrate farnesyl pyrophosphate (FPP) to form the specialized sesquiterpene skeletons [92]. In 2013, Lange and Turner proposed that the *X. strumarium* plant could biosynthesize xanthanolides in glandular cells and distribute them on the surface of plant organs [93]. Soetaert et al., conducted an RNA-sequencing strategy to elucidate the biosynthetic pathways of STLs and finally identified some key genes involved in STL biosynthesis [94]. In 2016, Zhang et al., successfully identified three *X. strumarium* sesquiterpene synthase genes, which provided the molecular basis for the biosynthesis of sesquiterpenoids in *X. strumarium* [95]. With more and more plant sesquiterpene synthases being isolated, engineering synthesis become possible to prepare xanthanolides in microorganisms [35]. However, due to the mechanism complexity of the synthases and the competitive pathways, the yield of biosynthesis was still at a low level, which limited the commercialization of these methods. To date, xanthanolides were still produced by plant extraction or chemical synthesis.

## 5. Conclusions

Xanthanolides were distinctive sesquiterpenoids exist in the genus *Xanthium*, which were responsible for the significant bioactivities of the *Xanthium* sepecies. Over 30 xanthanolides had been isolated from the species of *Xanthium* since the first discovery of xanthatin (**1**) and xanthinin (**14**), which have drawn much attentions in pharmacological research because of its broad activity and specific structures. Here in, the structures and biological activities of the xanthanolides in the genus *Xanthium* were summarized. Among all the xanthanolides, xanthatin, with α-methylene-γ-lactone ring as key active group, was recognized as a key molecule and had been studied in-depth against tumor and inflammatory diseases. Meanwhile, the total synthesis tactics of xanthanolides were also summarized for the efficient preparation of the xanthanolides. Above all, xanthanolides, as naturally occurring sesquiterpenoids, were thought to be promising in the medicinal industry and more studies should to be conducted for their potential application values.

## Data Availability

Not applicable.

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
