# Peer review of "Xanthanolides in Xanthium L.: Structures, Synthesis and Bioactivity"

_molecules, 2022, doi:10.3390/molecules27238136_

Round 1
Reviewer 1 Report
Zhang et al. have reviewed the chemistry and biology of the xanthanolides, an interesting class of plant secondary metabolites with various biological activities. There are a number of issues in the text which I have indicated in the attached annotated pdf but the coverage of the topic appears reasonable. I suggest inclusion of synthetic schemes instead of the current text-only version which is hard to grasp.
The authors should thoroughly check the italicization of binary species names – as far as I know, genera are not italicized instead but this may also depend on the journal policy.

Author Response
Dear reviewer 1
Re: Manuscript ID: molecules-1940031 and Title: Xanthanolides in Xanthium L.: structures, synthesis and bioactivity
Thank you for your letter and the reviewers’s comments concerning our manuscript entitled “Xanthanolides in Xanthium L.: structures, synthesis and bioactivity ” (molecules-1940031). Those comments are valuable and very helpful. We have read through comments carefully and have made corrections. Based on the instructions provided in your letter, we uploaded the file of the revised manuscript. Revisions in the text are shown using blue highlight for additions, and strikethrough font for deletions. The responses to the reviewer's comments are marked in red and presented following.
We would love to thank you for allowing us to resubmit a revised copy of the manuscript and we highly appreciate your time and consideration.
Sincerely.
Jiaojiao Zhang
Point 1:Synthetic schemes instead of the current text-only version which is hard to graspas as reviewers`s suggest.
Response:Thank you for the suggestion. We have added the schemes required as explained above (Scheme 1, page 9)
Point 2:The binary species name needs to be italicized for clarity.
Response:We apologize for the language problems in the original manuscript. The language presentation was improved with assistance from a native English speaker with appropriate research background.

Reviewer 2 Report
The review paper by Zhang and collaborators offers a brief yet effective overview of the main research works dedicated to the synthesis and bioactivities of xanthanolides from the Xanthium species. Overall, the manuscript is well-organized and has a clear structure. The table and the chart are useful and allow for a quick exploration of the literature results. The weakest point is the writing: the manuscript is riddled with typos and grammar mistakes, the concepts are sometimes difficult to understand, and, overall, the sentence structure is awkward. Also, verb tenses are used inconsistently throughout the paper (the past tense is sometimes used instead of the present and vice versa). Hence, the article should be carefully checked, possibly with the support of an English expert.
The authors surely cite many relevant works, but a quick search on SciFinder performed by this Reviewer returned many articles that were not included in the paper. Therefore, the following suggestions are offered:
· widen the search to include recent or overlooked articles pertaining to the topic
· add citations to relevant reviews (e.g., there are review works dedicated to specific areas of xanthanolide research, like synthesis, biosynthesis etc.)
· include a PRISMA diagram to identify the literature sources used for the review and clarify the process of article selection.
Moreover, the authors state that, despite the advances in xanthanolide synthesis, most of these molecules are still derived from plant extraction. Hence, this Reviewer would suggest expanding on this topic and include a paragraph or at least some references dedicated to the extraction of xanthanolides.
Finally, more specific and significant keywords should be added.
Author Response
Dear reviewer 2
Re: Manuscript ID: molecules-1940031 and Title: Xanthanolides in Xanthium L.: structures, synthesis and bioactivity
Thank you for your letter and the reviewers’ comments concerning our manuscript entitled “Xanthanolides in Xanthium L.: structures, synthesis and bioactivity ” (molecules-1940031). Those comments are valuable and very helpful. We have read through comments carefully and have made corrections. Based on the instructions provided in your letter, we uploaded the file of the revised manuscript. Revisions in the text are shown using blue highlight for additions, and strikethrough font for deletions. The responses to the reviewer's comments are marked in red and presented following.
We would love to thank you for allowing us to resubmit a revised copy of the manuscript and we highly appreciate your time and consideration.
Sincerely.
Jiaojiao Zhang
Point 1: The English language needs to be revised for clarity . (Including but not limited to spelling mistakes. Sentence errors and tense errors).
Response: We apologize for the language problems in the original manuscript. Following the reviewer’s suggestion .We have checked the manuscript very carefully again, and try our best to correct the non-standard writing which in the text are shown marked in blue .
Point 2: Widen the search to include recent or overlooked articles pertaining to the topic add citations to relevant reviews
Response:We deeply appreciate the reviewer’s suggestion. According to the reviewer’s comment,
We have changed some citations to adapt to the update of topics
.
Point 3: Include a PRISMA diagram to identify the literature sources used for the review and clarify the process of article selection.
Response:We appreciate the reviewer’s positive evaluation of our work, and our reply is as follows: We believe that adding diagrams reviewer’s suggested will change the content structure of the original text and affect the expression of the theme. In addition, many researchers have done relevant work on the specific analysis of the literature sources, so we did not add the diagram here.
Point 4: Reviewer would suggest expanding on this topic and include a paragraph or at least some references dedicated to the extraction of xanthanolides.
Response:Thank you for your comments, the discussion regarding this question is presented following:As for the extraction of xanthanolides, The corresponding extraction work has been mentioned in many literatures and the extraction process is more detailed. Therefore, we did not show it as a part of the article.

Round 2
Reviewer 2 Report
This Reviewer is satisfied with the corrections made to the manuscript and acknowledges the authors’ responses to Points 3 and 4. With regard to the content, the article can be deemed suitable for publication. However, despite the efforts made by the authors to improve the language and style were certainly appreciated, a much more radical revision of the English is necessary before the publication. This Reviewer recommends the support of a native speaker, or the use of the language editing services offered by MDPI.
Author Response
Dear reviewer 2
Re: Manuscript ID: molecules-1940031 and Title: Xanthanolides in Xanthium L.: structures, synthesis and bioactivity
Thank you for your letter and the reviewers’ comments concerning our manuscript entitled “Xanthanolides in Xanthium L.: structures, synthesis and bioactivity ” (molecules-1940031). Those comments are valuable and very helpful. We have read through comments carefully and have made corrections. Based on the instructions provided in your letter, we uploaded the file of the revised manuscript. Revisions in the text are shown using blue highlight for additions, and strikethrough font for deletions. The responses to the reviewer's comments are marked in red and presented following.
We would love to thank you for allowing us to resubmit a revised copy of the manuscript and we highly appreciate your time and consideration.
Sincerely.
Jiaojiao Zhang
